# Public Perceptions of the Role of Lifestyle Factors in Cancer Development: Results from the Spanish Onco-Barometer 2020

**DOI:** 10.3390/ijerph181910472

**Published:** 2021-10-05

**Authors:** Dafina Petrova, Josep Maria Borrás, Marina Pollán, Eloísa Bayo Lozano, David Vicente, José Juan Jiménez Moleón, Maria José Sánchez

**Affiliations:** 1CIBER of Epidemiology and Public Health (CIBERESP), 28029 Madrid, Spain; mpollan@isciii.es (M.P.); jjmoleon@ugr.es (J.J.J.M.); mariajose.sanchez.easp@juntadeandalucia.es (M.J.S.); 2Escuela Andaluza de Salud Pública (EASP), 18011 Granada, Spain; 3Instituto de Investigación Biosanitaria ibs.GRANADA, 18012 Granada, Spain; 4Department of Clinical Sciences, University of Barcelona, 08908 Barcelona, Spain; jmborras@iconcologia.net; 5Bellvitge Biomedical Research Institute, 08908 Hospitalet, Spain; 6NHS Cancer Strategy, Ministry of Health, 28046 Madrid, Spain; 7National Center for Epidemiology, Health Institute Carlos III, 28029 Madrid, Spain; 8University Hospital Virgen Macarena, 41009 Seville, Spain; eloisabayo@gmail.com (E.B.L.); david.vicente.sspa@juntadeandalucia.es (D.V.); 9Department of Preventive Medicine and Public Health, University of Granada, 18010 Granada, Spain

**Keywords:** cancer prevention, perceptions, lifestyle habits, risk factors, public knowledge, awareness, population-based survey

## Abstract

The European Code against Cancer recommends not to smoke, to avoid alcohol consumption, to eat a healthy diet, and maintain a healthy weight to prevent cancer. To what extent is the public aware of the influence of these lifestyle factors on cancer development? The goal of the current study was to describe the perceived influence of four lifestyle factors (tobacco, alcohol, diet, and weight) on cancer development in the general population and identify factors related to low perceptions of influence. We analyzed data from the 2020 Onco-barometer (n = 4769), a representative population-based survey conducted in Spain. With the exception of smoking, lifestyle factors were among those with the least perceived influence, more so among the demographic groups at higher risk from cancer including men and older individuals (65+ years). Individuals from lower socio-economic groups were more likely to report not knowing what influence lifestyle factors have on cancer. Lower perceived influence was also consistently related to perceiving very low risk from cancer. Overall, although there is variation in perceptions regarding the different lifestyle factors, low perceived influence clusters among those at higher risk for cancer. These results signal the need for public health campaigns and messages informing the public about the preventive potential of lifestyle factors beyond avoiding tobacco consumption.

## 1. Introduction

Cancer is the second leading cause of death in Europe, causing more than 250 deaths per 100,000 persons each year [1]. Although there is a genetic component to the disease, a large proportion of cancers have their roots in environmental and lifestyle factors and can thus be prevented [2]. The lifestyle factors with the strongest influence on cancer mortality are tobacco, alcohol consumption, diet (e.g., low fruit and vegetable intake), and overweight/obesity [3].

For instance, globally, tobacco use is estimated to be responsible for 81% of lung cancers in men and 58% in women [4]. Alcohol consumption is estimated to be globally responsible for 38% of oral cavity/pharynx cancers in men and 17% in women, and 29% of liver cancers in men and 13% in women [4]. Diet is responsible for many colorectal cancer cases. For instance, in the UK, 12% of cases are estimated to be preventable through the consumption of foods rich in fiber, 5% through lower consumption of red meat, and 10% through lower consumption of processed meat [5]. Finally, high body-mass index (BMI) is responsible for 12% of high-BMI-related cancers in men and 13% in women worldwide [6].

Overall, the World Health Organization estimates that one-third of deaths from cancer are caused by tobacco use, high body-mass index, alcohol consumption, low fruit and vegetable consumption, and lack of physical activity [7]. Because there is now sufficient evidence that these factors cause cancer, they have a central place in cancer prevention recommendations. The World Cancer Research Fund and the European Code against Cancer recommend not to smoke, to avoid alcohol consumption, to eat a healthy diet, be physically active, and maintain a healthy weight. It is estimated that adherence to these lifestyle recommendations and avoiding other risk factors such as certain infections and exposure to carcinogens and environmental pollution can prevent between 30 and 50% of all cancer cases [7]. Unfortunately, research from many countries shows that adherence in the general population is worryingly low [5,8,9,10,11].

Successful prevention requires both collective action (e.g., policies that limit risk exposure at the population level) and individual action (e.g., avoiding or reducing harmful exposures) [12]. For individual action to take place, the public needs to be aware of what the risk factors for cancer are. Unfortunately, population surveys in both higher and lower income countries have consistently shown that, with the exception of tobacco consumption, awareness of cancer risk factors is low [13,14,15,16,17,18,19,20,21,22].

In fact, among the different risk factors for cancer, awareness is frequently the lowest for lifestyle factors such as diet, weight, and alcohol consumption [14,15,16,18,20,23]. Awareness of lifestyle risk factors has been generally lower among men and persons with lower income or education [19,21,22,23,24,25]. This is especially important because these socio-demographic groups are at higher risk of cancer, probably at least partially due to higher exposure to diverse lifestyle risk factors [26].

Previous research has also shown that the public is generally more knowledgeable about the role of lifestyle factors (especially diet and physical activity) in cardiovascular disease compared to cancer [21]. However, people generally perceive more risk from cancer than from other chronic diseases [27]. Cancer is also perceived as more serious and evokes more fear than cardiovascular diseases, neurodegenerative diseases, AIDS, and mental disorders, among others [28,29]. Hence, informing the public about the importance of lifestyle factors in cancer development could help improve recommendation adherence because people may be especially motivated to prevent cancer.

In Europe, with a few exceptions [24], most population surveys measuring awareness of lifestyle risk factors for cancer were conducted in Northern and Western countries including the UK, Ireland, Denmark, and Sweden [15,20,21,23,30] about 10 years ago. More recent data from France and the US suggests that awareness of the role of certain lifestyle factors, especially diet, is slightly increasing [18,24], showing the need for updated evidence regarding the public perception of lifestyle risk factors for cancer.

In Spain, where the current study was conducted, perceptions of cancer risk factors in the general population were measured 10 years ago in the Onco-barometer 2010 survey of the Spanish Association against Cancer [13,31], showing perceptions similar to those found in other countries. In particular, among the 7938 respondents, tobacco was the factor with highest perceived importance for cancer development; weight was the factor with least perceived importance, whereas alcohol and diet occupied intermediate positions [13,31].

Recently, an update to the National Cancer Strategy of Spain was published [32], putting larger emphasis on the primary prevention of cancer through healthy lifestyle. One of the new objectives of the national strategy is to raise awareness of the European Code against Cancer and promote a healthier lifestyle in the population via interventions targeting diet, physical activity, and alcohol and tobacco consumption. Knowledge of the current public perceptions of the influence of lifestyle risk factors on cancer development could provide a useful baseline against which results of the cancer strategy can be evaluated and can serve to identify population segments with lower awareness.

Hence, the goals of the current study were to (1) describe the current perceived influence of diverse factors on cancer risk among the general population, with an emphasis on four lifestyle risk factors (weight, alcohol, diet, and tobacco) and (2) describe how this perceived influence varies as a function of socio-demographic factors, self-reported lifestyle, and perceived cancer risk.

## 2. Materials and Methods

We used data from the 2020 Spanish Onco-barometer, a periodic population-based survey conducted by the Spanish Association against Cancer, the leading NGO in Spain on cancer prevention, patient care, and research on cancer (www.aecc.es, accessed on 3 October 2021) [33]. The survey assesses knowledge and attitudes towards cancer and its previous edition was in 2010 [13,31].

Computer-assisted interviews were carried out by a specialized research market company under contract of the Spanish Association against Cancer. A two-stage sampling design was used to obtain a sample representative of the Spanish population. First, stratified random sampling proportional to the population sizes of the Spanish Autonomous Regions was used for household selection. Then, sampling units were selected by applying sex and age quotas with one interview per household. The distribution of mobiles/landlines was 50%/50%. Men and women, 18 years or older, who were able to speak Spanish were eligible.

The announcement of the state of emergency by the Spanish government and the associated restrictions due to the coronavirus pandemic caused the interruption of the original data collection plan which was renewed as soon as conditions allowed, generating two survey waves: the first wave between 10 February 2020 until 13 March 2020 and the second wave between 24 August 2020 until 08 September 2020.

No ethical approval was required for the current study because it was based on analysis of secondary data. The dataset used for this study can be requested from the Spanish Association against Cancer.

### 2.1. Measures

#### 2.1.1. Perceived Influence of Different Risk Factors in Cancer Development

This was based on the question “How much influence do you think each of the following aspects has for a person to develop cancer?” with answer options from 1 (has no influence) to 10 (lots of influence). The question was asked for 10 factors related to cancer, of which four were the lifestyle risk factors of interest for the current research: tobacco, alcohol, diet, and weight; and six were other additional factors: sunlight exposure, family history of cancer, atmospheric pollution, radiation, sexually transmitted diseases, and toxic substances. It was also recorded if respondents answered “I do not know” or did not respond to the question (these were not introduced as answer options to respondents).

#### 2.1.2. Demographic Characteristics

Data were collected on sex and age. Civil status was categorized into five groups: (1) single, (2) married or cohabiting with a partner, (3) separated or divorced (currently not cohabiting with a partner), (4) widowed, and (5) other. Socio-economic position was categorized into 7 groups following the methodology of the Spanish National Health Survey and the Spanish Epidemiology Society, based on information about education and income [34]). Respondents were asked if they had any personal history of cancer (yes vs. no) and if they had a close family member diagnosed with cancer (yes vs. no).

#### 2.1.3. Self-Reported Lifestyle

This was based on the question “Would you say your lifestyle is very healthy, healthy, somewhat unhealthy, or not healthy at all?”, with four answer options—“very healthy”, “healthy”, “somewhat unhealthy”, and “not healthy at all”.

#### 2.1.4. Perceived Risk from Cancer

This was based on the question “Do you think your risk of developing any type of cancer during your lifetime is very high, high, low, or very low?”, with the answer options “very high”, “high”, “low”, “very low” or “I do not know”.

### 2.2. Analyses

First, we described the perception of influence of the 10 risk factors using measures of central tendency and the percentage of “I do not know” responses. Not responding to the question was extremely rare (between 0.1% and 0.3%), so it was grouped with the “I do not know” responses. We then calculated the percentage of respondents who perceived a high influence of each factor on cancer development, defined as an importance rating above the psychological midpoint of the scale (>5) (“high influence”). The rest of the responses, including influence ratings ≤5 and the minority responses representing “I do not know” or not responding to the question, were assigned to the “low influence” category, because in one way or another, they were inconsistent with high perceived influence of the factors on cancer risk.

Second, using multiple logistic regressions, we investigated what variables were associated with “low influence” ratings of each lifestyle factor. We present this binary outcome variable as primary because it does not exclude respondents who answered with “I do not know”, and because it facilitates comparisons with most previous research that reports binary recognition of the risk factors in the population (e.g., awareness vs. unawareness). However, using analogous multiple linear regressions, we also analyzed the original continuous-influence ratings and compared the results. In this analysis, “I do not know” responses are excluded, but the whole range of responses is analyzed without using an arbitrary cutoff in order to define low and high perceived influence.

Finally, for each respondent the number of lifestyle factors with “low influence” rating was calculated (a score from 0 to 4). Using multiple Poisson regression, we then investigated what variables were associated with a higher number of lifestyle factors with low influence ratings. We used the number of factors with low instead of with high influence ratings, because the distribution of the latter (high influence) was negatively skewed and highly under-dispersed, whereas the distribution of the former (low influence) was better suited for this type of analysis.

Effects were considered as significant if the 95% confidence intervals excluded 1 for logistic and Poisson regressions and 0 for linear regressions, respectively. Analyses were conducted in R using the package *survey* (v. 3.37) [35]) and sampling weights were applied in all analyses. In the case of missing data, analyses were based on complete cases. We included wave (first or second) as a control variable in all analyses to adjust for possible differences between the two waves.

## 3. Results

The number of respondents was n = 4769 with a response rate of 64.1%. Demographic characteristics and other descriptive statistics of the sample are displayed in Table 1. Figure 1 displays the distributions of the influence ratings for all 10 risk factors, together with the percentage of “I do not know” responses. Table A1 reports in more detail the influence ratings for the four lifestyle factors as a function of demographic characteristics and the rest of the variables.

With the exception of tobacco, which was the factor with the largest perceived influence on cancer, the other lifestyle factors were rated among the least influential, only outranked by sexually transmitted diseases. Figure 1 shows that weight, alcohol, and diet were characterized by asymmetric distributions, where the mean ratings were notably lower than the medians, due to the presence of significant minorities giving low ratings that diverged from the majoritarian relatively high-influence perceptions. Whereas tobacco was given a high influence rating (>5) by 95.2% of respondents, weight was assigned such a rating by only 65.4%, followed by alcohol (78.7%) and diet (80.7%) (see Table 1).

### 3.1. Weight

The results of a multiple logistic regression with outcome low perceived influence of weight (1 = low influence; 0 = high influence) are displayed in Figure 2 (Panel A). However, there were no significant effects.

The analysis of the continuous ratings (see Figure A1, Panel A) showed that, considering only those respondents who assigned a rating, women (vs. men), those from the lowest socio-economic group (vs. the highest), those who participated in the second survey wave (vs. the first), and those who reported a “very healthy” lifestyle (vs. “healthy”), perceived a higher influence of weight on cancer development.

It is of note that the lowest socio-economic category had the highest percentage of “I do not know” responses (Table A1). A χ2 test indicated that the percentage of “I do not know” responses varied significantly among the socio-economic groups, χ2(5) = 17.8, *p* = 0.003. However, when these responses were excluded in the continuous ratings analysis, those from the lowest socio-economic group rated the influence of weight as higher on average than respondents from the highest socio-economic category.

### 3.2. Alcohol

The results of a multiple logistic regression with outcome low perceived influence of alcohol (1 = low influence; 0=high influence) are displayed in Figure 2 (Panel B). Alcohol was less likely to be perceived as having high influence on cancer development among men (74.7%) than among women (81.9%) and among single (77.6%) compared to married or cohabiting (79.4%) respondents (see also Table 2). Perceived risk from cancer was also strongly related to the influence ratings of alcohol, with those who perceived very low (compared to very high) risk from cancer being the least likely to give it a high influence rating (Table 2). Finally, the youngest respondents (18-24) were significantly more likely to give a high influence rating to alcohol compared to all other age groups (84.6% vs. <80% for the rest of the groups).

The analysis of the continuous ratings (see Figure A1, Panel B) revealed additional effects. Among respondents who reported an influence rating, those who belonged to lower socio-economic groups (Groups 4 to 7 vs. Group 1) perceived that alcohol had a higher influence on cancer development.

Again, the lower socio-economic groups had higher rates of “I do not know responses” regarding the influence of alcohol, χ2(5) = 11.2, *p* = 0.048 (Table A1). However, when these responses were excluded in the continuous ratings analysis, the lower socio-economic groups rated the influence of alcohol as higher on average than respondents from the highest socio-economic category.

### 3.3. Diet

The results of a multiple logistic regression with outcome low perceived influence of diet (1 = low influence; 0 = high influence) are displayed in Figure 2 (Panel C). Diet was less likely to be perceived as having high influence on cancer development among men, the oldest respondents (65+), the less-privileged socio-economic groups, those who had no family history of cancer, and those who reported a very unhealthy lifestyle or very low perceived risk from cancer (see Table 2). Differences were particularly pronounced for age: whereas 87.3% of individuals 25–34 years old gave a high influence rating to diet, this was the case for only 71.9% for individuals 65+.

The analysis of the continuous ratings (see Figure A1, Panel C) revealed similar trends, with the difference being that the effects of socio-economic position were not significant. Again, the rates of “I do not know responses” were higher among the less-privileged socio-economic groups, χ2(5) = 19.5, *p* = 0.002 (Table A1), which could explain why there were differences in the logistic regression analysis, where these responses are counted towards the “low influence” group.

### 3.4. Tobacco

The results of a multiple logistic regression with outcome low perceived influence of tobacco (1 = low influence; 0 = high influence) are displayed in Figure 2 (Panel D). Smoking was less likely to be perceived as having high influence on cancer development among the oldest respondents (65+), socio-economic groups 4 and 5 (self-employed and qualified technical occupations), those who had personal history of cancer, and those who reported a very low perceived risk from cancer (see Table 2).

The analysis of the continuous ratings (see Figure A1, Panel D) revealed similar trends, with the addition that single respondents perceived that tobacco had lower influence on cancer development than married respondents.

### 3.5. Number of Lifestyle Risk Factors with Low Perceived Influence

Only 1.6% of respondents gave a low influence rating to all four lifestyle factors; 6.4% gave to three, 14.9% to two, 24.8% to one, whereas 52.4% gave high influence ratings to all four lifestyle factors.

The results of a multiple Poisson regression with outcome the number of lifestyle risk factors with low perceived influence are displayed in Figure 3 and descriptive statistics are displayed in Table 2. Men gave low influence ratings to more lifestyle factors (Median = 1) compared to women (Median = 0) (Table 2). The number of lifestyle factors with low perceived influence also increased with age, with the two oldest age groups (55–64 and 65+) giving such a rating to a larger number of lifestyle factors (Median = 1 compared to Median = 0 for the rest of the groups). The difference between the 65+ group and the youngest group (18–24) was significant (see Figure 3). Respondents from the less-privileged socio-economic groups gave more low influence ratings compared to the most privileged group. Single respondents also gave more low influence ratings than married or cohabiting respondents. Finally, those who perceived their cancer risk to be very low (compared to very high) gave more low influence ratings.

## 4. Discussion

The results from the 2020 Spanish Onco-barometer survey reveal that public awareness of the influence of lifestyle factors on the development of cancer should be improved, especially in certain population segments. In particular, one in three Spaniards failed to recognize the high influence of weight on cancer risk and about one in five of alcohol and diet. In addition, with the exception of sexually transmitted diseases, lifestyle factors ranked as those with the least perceived influence by the population.

Population awareness of the link between healthy lifestyle and cancer is essential for cancer prevention. Although adherence to cancer prevention recommendations regarding diet seems to be somewhat higher in Spain compared to, for instance, Anglo-Saxon countries [36,37], there is still much room for improvement. These results suggest that low adherence could be partially due to low perception of influence of lifestyle factors on cancer risk among the general population.

Comparisons with the 2010 Spanish Onco-barometer survey show that the perceived influence of alcohol and smoking has remained similar, and smoking continues to be seen as the most influential risk factor for cancer in the Spanish population. A more positive finding is that we observe an increase in perceived influence by 1 point for diet and 1.5 points for weight (on the 1 to 10 scale of perceived influence) compared to the Onco-barometer data collected 10 years ago [31]. These results echo findings from the USA and France showing that knowledge about the role of overweight/obesity, diet, and other nutrition-related factors in cancer development is increasing, mostly due to awareness about red meat as a risk factor [18,24]. For instance, the AICR Cancer Awareness survey in the US shows that awareness of the link between overweight/obesity and cancer has risen from 35% in 2001 to 53% in 2019 [18]. However, still less than 50% of Americans understand the role diet plays in reducing cancer risk, with only 46% being aware of the risk associated with diets high in fat and 42% of diets low in fruit and vegetable consumption [18].

In the current study, the perceived influence of lifestyle factors was lowest among those demographic groups at higher risk of developing cancer including men and older individuals (e.g., 65+). These findings are in accordance with results from some previous studies that documented better knowledge of cancer risk factors among women [22,38] and younger people [15,38]. For instance, in the French Cancer Barometer, younger respondents identified more lifestyle risk factors, whereas older respondents were more likely to endorse psychosocial causes of cancer [39]. In the current study, older individuals were notably less likely to perceive diet as an influential factor, as were individuals who reported having an unhealthy lifestyle.

Multiple studies have found that people with lower education level or of lower socio-economic status are less aware of the relationship between cancer and diverse risk factors [19,21,22,23,24,30]. Lower knowledge of cancer risk and prevention factors could lead to lower adherence to risk-reducing behaviors and thus be one of the multifactorial mechanisms driving socio-economic disparities in cancer incidence and survival [26,40]. In the current study we found similar results regarding the perceived influence of lifestyle factors when we analyzed the influence ratings categorized into low vs. high groups. In particular, respondents from the less-privileged socio-economic groups gave fewer high influence ratings (>5) compared to the more privileged groups. This suggests that, similar to previous research and assuming that an influence rating >5 indicates more awareness, awareness of the influence of lifestyle factors on cancer risk may be lower among individuals from lower socio-economic backgrounds.

However, analyzing the continuous influence ratings and the “I do not know” responses separately (Table A1), we found intriguing results. In particular, the percentages of “I do not know” responses were higher among the lower socio-economic groups for three out of four lifestyle factors. However, among those lower socio-economic group respondents who did give a rating, in the case of weight and alcohol, it was on average higher than the ratings given by respondents from higher socio-economic groups. On one hand, this could reflect important differences in perceptions within lower socio-economic groups between those who have and have not received information or recommendations regarding the prevention of cancer (e.g., at a regular checkup with their primary care physician). On the other hand, it could also reflect a response bias, such as a tendency among lower socio-economic groups to give extreme responses due to lower understanding of the 10-point scale in the context of a telephone interview. Previous studies assessing awareness of cancer risk factors have used very diverse questions and have mostly reported results based on categorized responses (e.g., aware vs. unaware), (e.g., [15,18,30] an approach we also followed partially to facilitate comparisons. The current results suggest that, at least with regard to socio-economic differences, it would be important to investigate responses in more detail and/or using diverse questions and response scales. An analysis focused only on the continuous ratings and ignoring the “I do not know responses” would have produced only a partial understanding of how socio-economic status shapes perceptions.

A worrying finding from a public health perspective is the strong relationship between perceiving very low risk from cancer and lower perceived influence of alcohol, diet, and tobacco on cancer risk. This is problematic because it is possible that lack of knowledge contributes to a false perception of low risk from cancer and missed opportunities for cancer prevention where they are needed the most. To illustrate, both men and women who are not aware of alcohol as a risk factor for cancer are more likely to report excessive alcohol consumption [41]. Perceived risk is a central variable in many theoretical models aiming to predict health behavior and is an important driver of cancer preventive behaviors such as screening [42]. Individuals who perceive very low risk from cancer may also be less likely to follow cancer prevention guidelines. This information could be very useful for targeting preventive activities at groups where low cancer-risk perception is combined with high actual risk of cancer.

Limitations of this study include a possible selection bias due to survey non-response and lack of detail in the items used to measure the perceived influence of lifestyle factors. For instance, participants were asked about the role of “diet and types of foods consumed” in general, without inquiring about the role specific types of foods, and were asked about “weight” and not overweight or obesity more specifically. It would be important to investigate this in more detail because previous studies show that people often have misconceptions about what factors in the diet influence risk [20]. For instance, many people believe in fictitious cancer causes such as using microwave ovens or drinking water from plastic bottles [43]. Such beliefs can be detrimental for cancer prevention because they might offer a false sense of protection by avoiding the fictitious causes. On the other hand, it should be mentioned that the objective of the survey was to offer a broad perspective on the perceived influence of risk factors. More detailed studies should be conducted to understand the differences in perceived risk of different food components or dietary patterns.

Additional limitations include the lack of assessment of other important risk factors for cancer such as age, physical activity, and breastfeeding. In addition, respondents were asked about the perceived influence of these factors for cancer development. Different responses might have been obtained if respondents were asked about cancer prevention instead.

Finally, the cutoff used to define high vs. low influence (influence rating of 5) was arbitrary and was not based on the distribution of the data (e.g., a median split would have placed the cutoff between 7 and 9 for the different factors). This was because the goal was to identify what factors characterize individuals with the lowest perceptions of influence rather than compare those who perceive more vs. less influence, a goal achieved by the continuous ratings analysis.

## 5. Conclusions

Lifestyle factors including diet, weight, and alcohol consumption continue to be among those with least perceived influence on cancer risk among the population, unfortunately more so among demographic groups at higher risk from cancer such as men and older individuals (65+). Individuals from lower socio-economic groups are more likely to report not knowing what influence lifestyle risk factors have on cancer. Finally, individuals reporting very low perceived risk form cancer (and a very unhealthy lifestyle in the case of diet) also report lower perceived influence of lifestyle factors. Overall, although there is variation in perceptions regarding the different lifestyle factors, low perceived influence clusters among those at higher risk for cancer. Future research should address whether changing these perceptions translates into higher adherence to cancer prevention recommendations. So far, these results signal the need for public health campaigns and messages informing the public about the preventive potential of lifestyle factors, beyond avoiding tobacco consumption.

## Figures and Tables

**Figure 1 ijerph-18-10472-f001:**
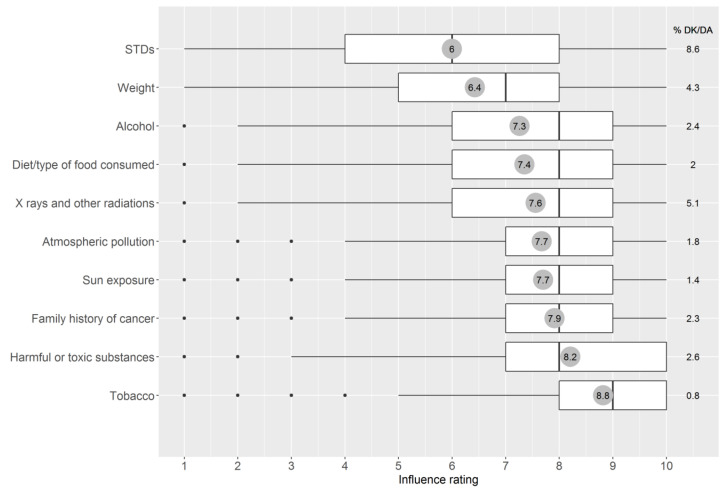
Influence ratings from 1 (has no influence) to 10 (lots of influence) for each of the 10 factors regarding its role in the development of cancer. Note: The line that divides each box in two is the median; the dimensions of the box are the interquartile range; the value in the grey circle is the mean. DK/DA = Percentage respondents indicating “I don’t know” or who do not answer. STDs = sexually transmitted diseases.

**Figure 2 ijerph-18-10472-f002:**
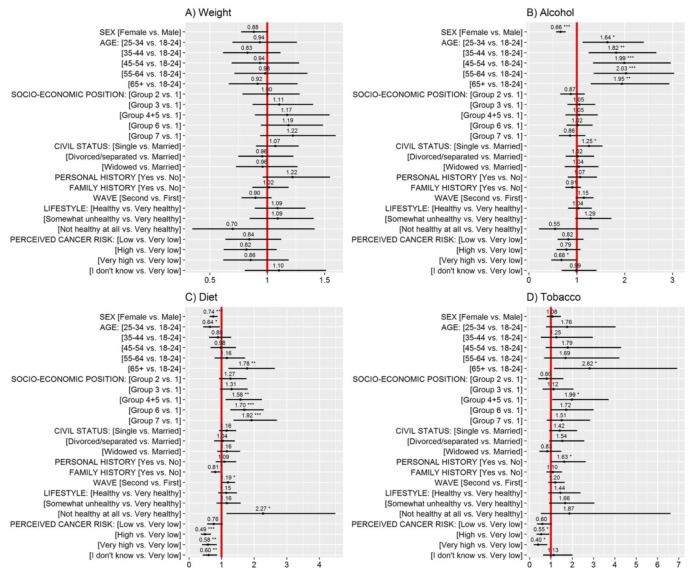
Odds ratios (OR, black dots) and 95% confidence intervals (CI) derived from multiple logistic regression models with outcomes influence ratings of (**A**) weight, (**B**) alcohol, (**C**) diet, and (**D**) tobacco on the development of cancer (1 = “low influence”; 0 = “high influence”). Note: * *p* < 0.05, ** *p* < 0.01, *** *p* < 0.001. If the OR and its CIs are to the left of the red reference line, the indicated group has a lower probability of assigning a “low influence” rating compared to the reference group. If the OR and its CIs are to the right of the red reference line, the indicated group has a higher probability of assigning a “low influence” rating to the risk factor compared to the reference group. For civil status, the category “Married” includes married and cohabiting.

**Figure 3 ijerph-18-10472-f003:**
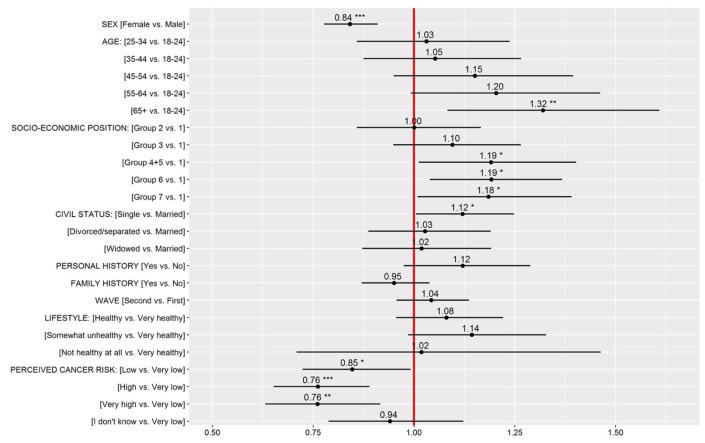
Relative score increases (black dots) and 95% confidence intervals (CI) derived from a Poisson regression model with outcome the number lifestyle risk factors with low perceived influence. Note: * *p* < 0.05, ** *p* < 0.01, *** *p* < 0.001. If the estimated effect and its CIs are to the left of the red reference line, the indicated group has a lower score (i.e., fewer factors with low perceived influence) compared to the reference group. If the estimated effect and its CIs are to the right of the red reference line, the indicated group has a higher score (i.e., more lifestyle factors with low perceived influence) compared to the reference group. For civil status, the category “Married” includes married and cohabiting.

**Table 1 ijerph-18-10472-t001:** Demographic characteristics and descriptive statistics of the sample (n = 4769).

Variable	Category	Percentage
Sex	Men	43.5
	Women	56.5
Age	18–24 years	8.3
	25–34 years	13.7
	35–44 years	19.2
	45–54 years	19.2
	55–64 years	15.7
	65+ years	23.9
Socio-economic position	Group 1 (highest)	11.4
	Group 2	15.3
	Group 3	17.9
	Group 4	1.4
	Group 5	8.3
	Group 6	25.1
	Group 7 (lowest)	11.3
	Missing	9.4
Civil status	Married or cohabiting	51.0
	Single	33.1
	Divorced or separated	8.3
	Widowed	7.2
	Other	0.4
Personal history of cancer	No	90.6
	Yes	9.2
	Missing	0.1
Close family member with cancer	No	25.7
	Yes	74.1
	Missing	0.2
Wave	First	68.7
	Second	31.3
Self-reported lifestyle	Very healthy	12.6
	Healthy	70.5
	Somewhat unhealthy	15.2
	Not healthy at all	1.2
	Missing	0.5
Perceived risk from cancer	Very low	6.0
	Low	27.1
	High	42.5
	Very high	11.4
	Does not know	11.6
	Missing	1.3
High perceived influence in cancer development (influence rating > 5)	Weight	65.4
	Alcohol	78.7
	Diet	80.7
	Tobacco	95.2
	STDs	53.4
	X-rays and other radiations	78.8
	Atmospheric pollution	85.5
	Sunlight exposure	85.7
	Family history of cancer	86.4
	Harmful or toxic substances	90.1

Note: Socio-economic position: Group 1. Directors and managers of establishments with 10 or more employees and professionals traditionally associated with university degrees. Group 2. Directors and managers of establishments with fewer than 10 employees and professionals traditionally associated with university degrees. Group 3. Intermediate occupations: employees of the administrative type and professionals supporting administrative management. Group 4. Freelancers/self-employed. Group 5. Supervisors and workers in qualified technical occupations. Group 6. Qualified workers of the primary sector and other semi-qualified workers. Group 7. Unskilled workers.

**Table 2 ijerph-18-10472-t002:** Influence ratings according to socio-demographic groups and other variables. Note: SD = standard deviation; Med. = Median. Values in the grey rows are *p*-values based on Wilcoxon or Kruskal–Wallis tests in the case of the number of lifestyle factors with low ratings and chi-square tests in the case of the variables designating the percentage of respondents giving high influence ratings (weight, alcohol, diet, and tobacco).

Variable	Category	N	Number of Lifestyle Factors with Low Influence Ratings			Percentage of Respondents Giving a High Influence Rating (>5) to Each Lifestyle Factor			
			Mean	SD	Med.	Weight	Alcohol	Diet	Tobacco
**Sex**	Men	2072	0.88	1.04	1	64.1	74.7	78.2	95.3
	Women	2697	0.74	0.99	0	66.3	81.9	82.6	95.0
			<0.001			0.127	<0.001	<0.001	0.693
**Age**	18–24 years	394	0.74	0.96	0	63.6	84.6	80.8	96.7
	25–34 years	655	0.71	0.93	0	66.2	79.3	87.3	95.9
	35–44 years	913	0.74	1.00	0	67.5	77.9	83.9	96.7
	45–54 years	918	0.78	1.00	0	65.2	78.4	83.1	95.5
	55–64 years	751	0.83	1.01	1	63.3	77.6	81.1	95.2
	65+ years	1138	0.92	1.09	1	65.1	78.0	71.9	92.7
			<0.001			0.579	0.106	<0.001	<0.001
**Socio-economic position**	Group 1 (highest)	543	0.71	0.95	0	67.9	78.5	86.3	96.4
	Group 2	729	0.69	0.96	0	68.2	81.3	83.7	97.4
	Group 3	853	0.77	0.97	0	66.1	77.8	83.4	96.0
	Group 4	459	0.88	1.08	1	60.4	82.2	80.6	94.3
	Group 5	1198	0.85	1.06	0	65.1	76.0	77.1	92.3
	Group 6	540	0.86	1.04	1	63.8	78.1	78.5	94.1
	Group 7 (lowest)	2072	0.88	1.04	1	63.4	80.6	75.8	94.2
			0.004			0.304	0.341	<0.001	0.001
**Civil status**	Married/cohabiting	2433	0.78	1.01	0	66.0	79.4	81.1	95.4
	Single	1577	0.81	1.02	0	64.4	77.6	81.8	95.4
	Divorced/separated	394	0.79	0.99	0	65.5	79.8	81.2	94.3
	Widowed	344	0.88	1.03	1	65.8	78.9	73.3	94.3
			0.305			0.770	0.564	0.003	0.701
**Personal history of cancer**	No	4322	0.79	1.01	0	65.6	78.9	81.1	95.4
	Yes	440	0.90	1.07	1	62.0	76.9	77.4	93.1
			0.031			0.127	0.326	0.072	0.041
**Close family member with cancer**	No	1226	0.86	1.05	0	65.0	77.3	76.7	95.0
	Yes	3533	0.78	1.00	0	65.5	79.3	82.2	95.2
			0.037			0.784	0.165	<0.001	0.778
**Survey wave**	First	3278	0.79	0.99	0	64.4	79.5	81.6	95.4
	Second	1491	0.82	1.06	0	67.3	77.1	78.5	94.7
			0.788			0.053	0.065	0.011	0.336
**Self-reported lifestyle**	Very healthy	602	0.76	1.00	0	66.7	79.3	81.5	96.1
	Healthy	3362	0.80	1.02	0	65.3	79.0	80.5	95.1
	Somewhat unhealthy	723	0.82	1.00	0	64.6	76.6	81.9	95.0
	Not healthy at all	57	0.72	0.82	1	71.2	90.1	72.4	93.6
			0.693			0.710	0.083	0.315	0.689
**Perceived risk from cancer**	Very low	288	1.01	1.13	1	61.1	75.1	70.0	92.7
	Low	1291	0.82	1.00	0	65.5	78.7	77.6	95.6
	High	2027	0.72	0.97	0	66.9	79.9	84.8	96.0
	Very high	545	0.75	0.95	0	65.3	81.7	81.6	96.7
	Does not know	554	0.96	1.14	1	60.2	74.5	78.2	91.5
			<0.001			0.023	0.012	<0.001	<0.001

## Data Availability

The dataset used for the current study can be requested from the Spanish Association against Cancer (Asociación Española contra el Cáncer: www.aecc.es, 3 October 2021).

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
