# Peer review of "Public Perceptions of the Role of Lifestyle Factors in Cancer Development: Results from the Spanish Onco-Barometer 2020"

_ijerph, 2021, doi:10.3390/ijerph181910472_

Round 1
Reviewer 1 Report
Comments to ijerph-1384534
This manuscript is a very interesting and important study that reported the public perceptions of the role of lifestyle factors (four main factors tobacco, alcohol, diet and weight) in cancer development. The data were derived from the updated Spanish Onco-barometer 2020, in comparison with that of 2010. Therefore, the results have representative meaning, which inform government or official mission need to implement public health campaigns and disseminate knowledge to the public about the preventive potential of lifestyle factors beyond avoiding tobacco consumption.
Only one suggestion, the primary headline of “section 3” is missing.
Author Response
This manuscript is a very interesting and important study that reported the public perceptions of the role of lifestyle factors (four main factors tobacco, alcohol, diet and weight) in cancer development. The data were derived from the updated Spanish Onco-barometer 2020, in comparison with that of 2010. Therefore, the results have representative meaning, which inform government or official mission need to implement public health campaigns and disseminate knowledge to the public about the preventive potential of lifestyle factors beyond avoiding tobacco consumption.
Only one suggestion, the primary headline of “section 3” is missing.
A: Thank you for the positive feedback on the manuscript. We have corrected the section headings as requested.
Reviewer 2 Report
Thank you for the opportunity to review this interesting and well-written manuscript. I have a few comments below:
- Introduction, page 1, lines 11-48. It would be helpful to state the relevant population/s for the findings related to attributable cancers (e.g. tobacco responsible for 81% of lung cancers in men) - for example, what country do these findings relate to? These estimates vary markedly by population, and thus some context is needed here.
- Introduction, line 53. Does the one-third of preventable cancers relate only to these 5 recommendations, or to the wider set of recommendations (WCRF notes a total of 10 recommendations, for example).
- A little more detail on the results of the 2010 Onco-barometer survey could be added to the Introduction to provide more background and context to the current findings.
- Methods, line 132. Could 'civil status' be defined here for non-European readers? Another term commonly used might be 'marital status'.
- How does the sample compare to the general population? Is it generally representative in terms of the demographic factors measured?
- Table 2. Could p values be added to the table to show where significant differences lie?
- Discussion, line 379. Is there any evidence that "don't know" responses equate lower awareness? While this is probably a reasonable assumption, there is a need to be careful with wording here, as it may be that "don't know" responses are not correlated with awareness, or correlated in another way. Thus I'm not sure that it is wholly correct, for example, to conclude that ignoring the "don't know" responses would have "produced a misleading impression that awareness is higher in lower socio-economic groups" (lines 395-396).
Author Response
Thank you for the opportunity to review this interesting and well-written manuscript. I have a few comments below:
Introduction, page 1, lines 11-48. It would be helpful to state the relevant population/s for the findings related to attributable cancers (e.g. tobacco responsible for 81% of lung cancers in men) - for example, what country do these findings relate to? These estimates vary markedly by population, and thus some context is needed here.
A: It is indeed important to put these estimates in their proper context. We have now added this information to the text. In particular, the figures regarding tobacco, alcohol and BMI are worldwide estimations, whereas the figures relating to diet refer to the UK. This has now been explicitly mentioned.
Introduction, line 53. Does the one-third of preventable cancers relate only to these 5 recommendations, or to the wider set of recommendations (WCRF notes a total of 10 recommendations, for example).
A: It indeed relates to a wider set of recommendations. We have now updated this section to be more transparent (see page 2).
A little more detail on the results of the 2010 Onco-barometer survey could be added to the Introduction to provide more background and context to the current findings.
A: Thank you for this suggestion, we have added some further information about the results of the previous Oncobarometer 2010 on page 2.
Methods, line 132. Could 'civil status' be defined here for non-European readers? Another term commonly used might be 'marital status'.
A: We have added the requested information to the section on demographic characteristics on page 3.
How does the sample compare to the general population? Is it generally representative in terms of the demographic factors measured?
A: The sample is designed to be representative of the population based on residence (Autonomous Community) and age and sex distribution of the population. Sampling weights have been applied in all analyses to correct for any under or over sampling and produce estimates representative of the population.
Table 2. Could p values be added to the table to show where significant differences lie?
A: The p-values have been added to Table 2 as requested. In the case of Figures 2 and 3, the presence of a star next to the coefficient designates a p-value below the threshold for significance for each respective group.
Discussion, line 379. Is there any evidence that "don't know" responses equate lower awareness? While this is probably a reasonable assumption, there is a need to be careful with wording here, as it may be that "don't know" responses are not correlated with awareness, or correlated in another way. Thus I'm not sure that it is wholly correct, for example, to conclude that ignoring the "don't know" responses would have "produced a misleading impression that awareness is higher in lower socio-economic groups" (lines 395-396).
A: Thank you for this reflection. We agree that equating the “I don’t know” responses with low awareness can be premature on this occasion. It would also depend on how awareness is defined and measured, which can vary greatly across previous publications. We have thus changed the wording of this sentence to:
“An analysis focused only on the continuous ratings and ignoring the “I don’t know responses” would have produced only a partial understanding of how socio-economic status shapes perceptions.”
Thank you for the constructive comments on the manuscript.
This manuscript is a resubmission of an earlier submission. The following is a list of the peer review reports and author responses from that submission.
Round 1
Reviewer 1 Report
The authors have reported data from a national survey conducted by a research marketing group which was funded by a non-government organization. The language of the article also appears to be written by a marketing group; the following terms and phrases must be omitted from the text: unfortunately, clearly, for instance, …is responsible for…, worryingly, actually, probably at least partially, especially, and …showing the need...
While public perceptions of the role of lifestyle factors in cancer development is a very important question to answer, the survey questions used to answer this question do not appear to be validated, nor is the choice of analytical method scientifically sound. To elaborate, responses to the question “How much influence do you think each of the following aspects has for a person to develop cancer?” were reported in the non-peer reviewed report ten years ago, but there is no evidence for the methods of validating this question or its response scale, and how that correlates with actually recognizing that item as a factor. An importance rating greater than five is 100% arbitrary until validated. Thus, if the authors cannot cite where the question has been validated and the cutoff for importance rating equates recognition, this is nothing more than a marketing survey and should not be considered as a scientific study or endeavor.
Reviewer 2 Report
Main points
My main query regarding this paper is why the central participant question about how much influence a lifestyle factor has on developing cancer (line 118) is being conflated with awareness of that factor. The degree of influence and awareness are two separate concepts. Awareness is best quantified with open-ended questions such as "What do you think are the things that cause a person to develop cancer?" It is very difficult for a member of the public to give an estimate of the relative influence of a lifestyle factor.
Similarly in line 126, why is a respondent when asked how much influence a factor has for developing cancer, and responding that they do not know the answer, then categorised as not recognizing the factor? Knowing about relative influence and being aware seem to me to be two separate issues. Please justify. Also why was the factor designated as recognized if the respondent indicated an importance rating above the midpoint of the scale (>5). How was this value decided?
Other points
In the Introduction, it would be helpful if the authors gave statistics of the relative influences of lifestyle factors on overall cancer incidence - rather than for selected cancers - since this is what is being asked of participants in this study.
The paper focuses on instances where influence of a lifestyle factor is considered to be low. But it would be worth also mentioning examples where there is greater awareness. For example, the "not healthy at all" group had the highest awareness of alcohol as a risk factor (Table 2). Also Table 2 indicates that there was little difference between socioeconomic groups recognising weight, alcohol and tobacco as a risk factor. This seems to be contrary to some previous findings by others (And the authors state (line 291) that socioeconomic difference in recognition of factor is consistent.)
Related to this, please state what parameters were used to assign participants to the different socioeconomic groups.
line 121 Participants were asked about alcohol consumption whereas excessive alcohol consumption is usually specified in questionnaires since participants may have different perceptions of alcohol consumption. Please comment.
line 317 Saying that links between additives and cancer are fictitious ignores an increasing body of evidence for these links eg with titanium dioxide and emulsifiers
In the Discussion, it would be helpful if the authors based on their experience suggested ideas about raising awareness of lifestyle factors in cancer risk.
lines 28 and 334 'form' is mispelt